# Inhibition of SARS-CoV-2 Viral Channel Activity Using FDA-Approved Channel Modulators Independent of Variants

**DOI:** 10.3390/biom12111673

**Published:** 2022-11-11

**Authors:** Han-Gang Yu, Gina Sizemore, Ivan Martinez, Peter Perrotta

**Affiliations:** 1Department of Physiology and Pharmacology, School of Medicine, West Virginia University, Morgantown, WV 26506, USA; 2Clinical Medicine Resources, EZCARE Walk-in Medical Center, Moorefield, WV 26836, USA; 3Department of Microbiology, Immunology, & Cell Biology, Cancer Institute, School of Medicine, West Virginia University, Morgantown, WV 26506, USA; 4Anatomy & Laboratory Medicine, Department of Pathology, School of Medicine, West Virginia University, Morgantown, WV 26506, USA

**Keywords:** SARS-CoV-2, envelope, Orf3a, viroporin, channel, FDA-approved, blocker, viral replication

## Abstract

Background: SARS-CoV-2 has undergone mutations, yielding clinically relevant variants. Hypothesis: We hypothesized that in SARS-CoV-2, two highly conserved Orf3a and E channels directly related to the virus replication were a target for the detection and inhibition of the viral replication, independent of the variant, using FDA-approved ion channel modulators. Methods: A combination of a fluorescence potassium ion assay with channel modulators was developed to detect SARS-CoV-2 Orf3a/E channel activity. Two FDA-approved drugs, amantadine (an antiviral) and amitriptyline (an antidepressant), which are ion channel blockers, were tested as to whether they inhibited Orf3a/E channel activity in isolated virus variants and in nasal swab samples from COVID-19 patients. The variants were confirmed by PCR sequencing. Results: In isolated SARS-CoV-2 Alpha, Beta, and Delta variants, the channel activity of Orf3a/E was detected and inhibited by emodin and gliclazide (IC_50_ = 0.42 mM). In the Delta swab samples, amitriptyline and amantadine inhibited the channel activity of viral proteins, with IC_50_ values of 0.73 mM and 1.11 mM, respectively. In the Omicron swab samples, amitriptyline inhibited the channel activity, with an IC_50_ of 0.76 mM. Conclusions: We developed an efficient method to screen FDA-approved ion channel modulators that could be repurposed to detect and inhibit SARS-CoV-2 viral replication, independent of variants.

## 1. Introduction

SARS-CoV-2 has undergone continuous mutations throughout the pandemic, yielding clinically relevant variants such as Delta and Omicron. At the time of writing, the dominant variant was BA.5; this accounted for nearly 90% of all Omicron variants (including BA.2.12.1 and BA.4) in the US, according to the Centers for Disease Control and Prevention (CDC) [1]. Genomic mutations, especially those occurring in spike and in nucleocapsid proteins, have increased the ability of viruses to escape protection from antibodies developed from previous infections and/or vaccinations [2,3,4,5].

Currently, the antiviral drug Paxlovid (Pfizer) is the standard therapy to treat mild–moderate COVID-19 symptoms in patients with a confirmed viral infection [6]. The effective component of Paxlovid, nirmatrelvir, is a protease inhibitor [6]. Another antiviral drug, bebtelovimab (Eli Lily), is a neutralizing IgG1 monoclonal antibody that targets the spike protein that is effective against Omicron subvariants [7,8]. Nevertheless, Omicron variants are still causing around 400 deaths per day [9] [CDC data], serving as a reminder that efforts to produce effective therapies against SARS-CoV-2 variants are needed to help control the virus spread and to decrease the risk of infected patients who are at risk of progressing to severe symptoms.

There are two viroporins within the SARS-CoV-2 genome, Orf3a and E; both of which can form non-voltage-gated cation channels [10,11,12]. The structural protein envelope (E) and the accessory protein Orf3a are viroporins that consist of conserved channel activity that is not affected by massive mutations. This conserved channel activity also provides a target for the functional screening of FDA-approved ion channel modulators already used to treat other diseases. Orf3a and E channels are Na^+^ and K^+^ cation permeable: the Orf3a channel is more selective to K^+^ [13,14] whereas the E channel is more selective to Na^+^ (10 times higher than K^+^) ions [13]. The E channel is also permeable to Ca^2+^ ions [15] and its channel activity is influenced by pH values [16]. We previously reported that by using a combination of ion channel modulators and a fluorescent K^+^ ion assay, a rapid method, the Orf3a/E channel activity could be detected, a strong indicator of live SARS-CoV-2 in the blood of COVID-19 patients [17].

In this work, we report that clinically relevant mutations occurring in several variants of concern (VOC) did not affect Orf3a/E channel activity. Combined with our previous detection methods, we show that FDA-approved ion channel modulators can be screened for their ability to inhibit Orf3a/E channel activity. As Orf3a/E channel activity is directly associated with viral replication, we believe that repurposing ion channel modulators that are already clinically used for other diseases represents an effective alternative therapy for COVID-19 patients, independent of the causative variants.

## 2. Materials and Methods

### 2.1. COVID-19 Patient Blood and Nasal Swab Samples

The use of blood and nasal swab samples obtained from COVID-19 patients was approved by the West Virginia University Institutional Review Board. The research was discussed with each patient and if the patient consented to the research project, whole blood was drawn by a standard venipuncture into one tube containing EDTA for anticoagulation. A total of 12 blood samples from 12 patients were included. A total of 58 nasal swab samples were collected from COVID-19 patients. Twenty-one (21) Delta and thirty-seven (37) Omicron nasal swab samples were used. The SARS-CoV-2 variant status (Delta or Omicron) was verified by sequencing.

### 2.2. UV-Inactivated SARS-CoV-2 Variants

Isolated SARS-CoV-2 variants were obtained from the CDC. All purified viruses were UV-inactivated in a Biosafety Level 3 facility at the Health Sciences Center of West Virginia University. UV-inactivated viruses were studied in a Biosafety Level 2 facility.

### 2.3. Fluorescence K^+^ Assay and Orf3a/E Channel Activity Detection

A fluorescence-based potassium ion channel assay utilizes the ability of thallium (Tl^+^) to permeate K^+^ channels [18]. Once the K^+^ channels are open, Tl^+^ in the extracellular solution flows down its concentration gradient into the cells via K^+^ channels. Inside the cells, Tl^+^ binds to and activates a fluorogenic indicator dye preloaded into the cells, resulting in a dramatic increase in the fluorescence signal. This technique allows the rapid determination of K^+^ channel activity in a high-throughput manner [19]. We used a commercial kit (FluxOR Potassium ion channel assay, cat#: F10016, Thermo Fisher Scientific, Pittsburgh, PA, USA) according to the manufacturer’s instructions to study the fluorescence detection of the Orf3a/E channel activity.

Samples of 10 μL each were added to a 96-well plate. Each sample was duplicated in two separate wells. The first replicate contained only the assay solution to establish the baseline signal and the second contained both the assay solution and Orf3a/E channel blockers. The fluorescence of each sample was repeated 3–6 times using a BioTek Synergy H4 Hybrid Microplate Reader (BioTek Instruments Inc. Winooski, VT, USA).

### 2.4. Ion Channel Modulators

Emodin, gliclazide, and 4-aminopyridine (4-AP) were purchased from Sigma. A stock solution (50 mM) was prepared in DMSO. Approximately 0.5 mM of emodin and gliclazide were used in ~100 μL solution. DMSO at ~1–2% of the test solution had no effect on the test results.

Amantadine (ATD) and amitriptyline (ATT) were purchased from Thermo Fisher Scientific (Cat # 18-600-501) (Pittsburgh, PA, USA) and VWR (Cat # TCA0908-025G) (Radnor, PA, USA), respectively. Stock solutions (50 mM) for both ATD and ATT were prepared in H_2_O.

### 2.5. Antigen Testing of Blood Samples

Antigen testing was performed in a clinic laboratory using the CareStart Rapid Diagnostic Test for the detection of the SARS-CoV-2 antigen (Access Bio, Somerset, NJ, USA). Briefly, a nasopharyngeal swab was removed from a pouch and the swab was introduced into the nasal passage until it reached the posterior nasopharynx. The swab was rotated 3–5 times over the posterior nasopharynx, then removed from the nostril with a rotation to sample the anterior nares. In the laboratory, the seal was removed from an extraction vial containing the extraction buffer. The swab was placed in the extraction vial and vigorously rotated 5 times. The extraction vial was then squeezed whilst the swab was removed by rotating against the sides of the extraction vial to remove any excess fluid from the swab and a cap was placed on the extraction vial. The sample was mixed by tapping the bottom of the extraction tube whilst inverted and 3 drops were squeezed into the sample well. The results were read after ten minutes.

A red control line appeared at the top of the well next to the letter “C”. If the test was positive, a blue line appeared below the red line and across from the letter “T”. Positive results produced both red and blue lines. Negative results revealed only a single red line. A result with a blue line without a red line was invalid and test was repeated.

### 2.6. Data Analysis

Fluorescence data were collected using Gen5 2.0 microplate reader software (from BioTek), processed in Excel, and analyzed and plotted using GraphPad Prism 8.

For the IC_50_ calculation, four different concentrations (for example, 0 mM, 1.0 mM, 1.5 mM, and 4 mM) of one drug were studied. The IC_50_ was calculated using a non-linear fit for an inhibitor versus a normalized response model; specifically, Y = 100/(1 + X/IC_50_), where X was the concentration of the inhibitor and Y was the normalized fluorescence signal.

## 3. Results

Most mutations occur in the spike proteins, but a few occur in other proteins, including Orf3a and envelope proteins. Table 1 shows the mutations in the Orf3a and E channel proteins in the variants of concern (VOC).

We first verified the methodology we used to screen the drugs in the blood samples. Figure 1 shows the typical results for the detection of the Orf3a (red circle) and E (black diamond) channel activities. Emodin (hollow circle) and gliclazide were used to block the Orf3a and E channel activities, respectively. 4-AP was used to trigger a fluorescence response in the presence of the channel activity. As a negative control, normal plasma (NP) not infected with the virus was used, showing a lack of response to all the drugs studied (4-AP, emodin, and gliclazide).

We then tested the inhibitory effects of emodin and gliclazide in the isolated SARS-CoV-2 variants. Figure 2 shows the representative results using Orf3a/E channel blockers on isolated original (A), Alpha (B), Beta (C), and Delta (D) virus variants. At a combinational concentration of 2–4 mM, emodin and gliclazide could block the Orf3a and E channel activities in each of the SARS-CoV-2 variants studied.

We then used nasal swabs to test the inhibitory effects of emodin and gliclazide on the Orf3a/E channel activities. Figure 3 shows the results from the nasal samples prior to the Delta wave (A), during the Delta wave (B), and during the Omicron wave (C). Figure 3D shows a negative control of a swab sample for a negative PCR test. The Orf3a/E channel activities could be detected and blocked by emodin/gliclazide in all swab samples harboring the different variants.

To determine which FDA-approved ion channel blockers might inhibit the Orf3a/E channel activities, we used swab samples that were relatively easy to obtain. As shown in Figure 4, two clinically used drugs, the antiviral amantadine (ATD) and the antidepressant amitriptyline (ATT), could each inhibit the channel activity of the isolated SARS-CoV-2 Delta variant (A) and Delta swab samples (B—ATD, C—ATT) in a concentration-dependent manner.

Figure 5 shows the inhibitory effects of amitriptyline on the channel activity of a swab sample taken from a patient infected with the Omicron variant. ATT at 3 mM reduced the channel activity (red), which was further blocked by 1 mM emodin/gliclazide (A), indicating that the channel activity was from the Orf3a/E ion channels. Figure 5B shows the lack of effect by DMSO as a negative control (red) and emodin/gliclazide as a positive control (blue). We further tested different concentrations of ATT on the Omicron swab samples and determined that ATT inhibited the channel activity, with an IC_50_ of 0.76 mM.

Finally, we found, using blood samples, that the Orf3a/E channel activity was much weaker for Omicron compared with the parental (prior to Delta) and Delta variants (Figure 6), suggesting that a less severe pathology was generally associated with the Omicron variant than previous variants.

## 4. Discussion

SARS-CoV-2 variants that are associated with increased transmission rates most often express changes in spike proteins [27]. Whilst the main concern is that these mutations reduce the efficacy of a few vaccines [28] due to decreasing neutralizing activity and immune escape mechanisms [27,29], there is also a risk that a few novel variants may not be detected by current PCR- and antigen-based testing. Most recently, a new B.1.616 variant has been identified in which only 15% of patients infected with this variant were detected by RT-PCR [26], probably due to the distinct set of mutations and one deletion on the spike-1 N-terminal domain [30].

The FDA has recently emphasized the impact of further viral mutations (mostly by BA.4/BA.5) on SARS-CoV-2 antigen-based and molecular tests [31,32] because mutations are found in the nucleocapsid proteins (P13 L, DEL31/33, P203 K, G204 R, and S413 R) [23] that are targeted by many of these tests.

The NIH COVID-19 treatment guidelines [33] recommend four therapeutic drugs—Paxlovid, bebtelovimab, remdesivir, and molnupiravir—to treat non-hospitalized symptomatic adults testing positive for COVID-19. Paxlovid is a combination of two generic drugs, nirmatrelvir (which disrupts viral replication) and ritonavir (which enhances the effectiveness of nirmatrelvir by slowing its metabolization by the liver) [6,34]. Recent in vitro studies suggest that Paxlovid-resistant strains of the virus might arise [35,36]. Bebtelovimab is a neutralizing immunoglobulin G1 (IgG1) monoclonal antibody that binds to the spike protein [37]. Remdesivir is an antagonist of viral RNA-dependent RNA polymerase (RdRp), which inhibits SARS-CoV-2 replication [38]. Molnupiravir is a prodrug of the synthetic nucleoside derivative N4-hydroxycytidine, which introduces copy errors during viral RNA replication [39].

Additionally, the FDA has authorized Evusheld (AstraZeneca) for the preventive treatment of COVID-19 in patients with an immune compromise or a severe allergy [40]. Evusheld contains two recombinant human IgG1κ monoclonal antibodies, tixagevimab and cilgavimab; both of which bind to the distinct regions of the receptor-binding domain of the SARS-CoV-2 spike protein. A recent study showed that the protective effects of Evusheld were evaded by the Omicron BA.4.6 variant [41].

In this work, we targeted virally-encoded viroporins in SARS-CoV-2 genomes due to their roles in inflammation, viral replication, and pathogenesis [42,43,44]. ORF3a promotes virus release, viral replication, and virulence [26,45]. It can also induce apoptosis in host cells [46]. By interacting with the host immune system, ORF3a activates the pro-IL-1β gene expression and IL-1β secretion as well as NF-κB signaling, thus promoting the generation of cytokine storms [47,48].

The E protein deletion of SARS-CoV-2 causes a significant attenuation of viral virulence, suggesting an important role of E protein in the viral replication and disease pathogenesis [26]. SARS-CoV-2 E channel activity can cause injury to the lungs and heart. An injection of purified SARS-CoV-2 E protein into mice caused acute respiratory distress syndrome-like lung pathological damage [49]. E channel activity also disrupted the conduction of action potentials and Ca^2+^ homeostasis in cardiomyocytes, increasing the risk of arrhythmias [50].

In a pilot study, we tested two FDA-approved drugs associated with ion channel activity, amantadine and amitriptyline. Amantadine is used to treat movement difficulties in Parkinson patients [51,52]. It is also an antiviral drug that binds to the M2 proton channels in the influenza A virus [53,54]. The proton channels have a low permeability for other ions [55], which may explain the lower potency compared with amitriptyline.

Amitriptyline is a tricyclic antidepressant used to treat major depressive disorders [56] and to alleviate pain in diabetic neuropathy [57]. Amitriptyline inhibits the serotonin and noradrenaline uptake [58] and antagonizes 5-HT2 receptors that inwardly conduct the Na^+^ current [59]. This may explain the higher potency of amitriptyline than amantadine in the inhibition of the Orf3a/E channel activity.

It is worth noting that in the early exploratory experiments, we used high concentrations of drugs to shorten the experimental time and focus on testing the central idea, which was whether FDA-approved ion channel modulators used in a clinic could be repurposed to inhibit SARS-CoV-2 viral activity, directly controlled by the 3a/E channel activity.

Many Na^+^ and K^+^ channel modulators have been used in clinics to treat a variety of diseases such as cardiac arrhythmias and neuropathic pain [60,61,62,63,64,65]. Repurposing a few of these cation channel modulators that have inhibitory effects on Orf3a/E channel activity may represent an efficient approach to improve the therapy of COVID-19 patients whilst controlling the virus spread.

### Limitations of the Work

The sample size was not large in this work. The samples were collected when the disease prevalence was high (approximately up to 50%) at an outpatient clinic. A larger sample size is needed to test our method for detecting functional viral proteins in the blood during periods of lower disease prevalence. Additionally, we did not test the recent Omicron variants (BA.2–5).

## 5. Conclusions

We developed an efficient method to screen FDA-approved ion channel modulators that could be repurposed to inhibit SARS-CoV-2 viral replication, independent of the variants.

## Figures and Tables

**Figure 1 biomolecules-12-01673-f001:**
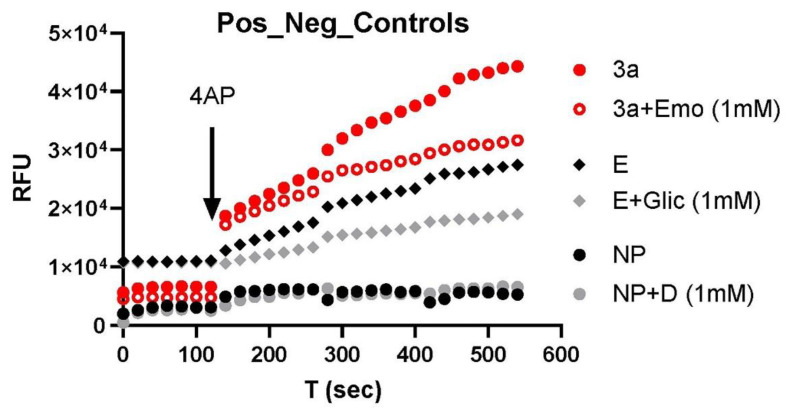
Channel activities of Orf3a and E inhibition by respective blockers: emodin, Orf3a channel blocker, gliclazide, and E channel blocker. NP: normal plasma. D: drugs (gliclazide + emodin). RFU: random fluorescence unit. The results were repeated in an additional 3–6 samples.

**Figure 2 biomolecules-12-01673-f002:**
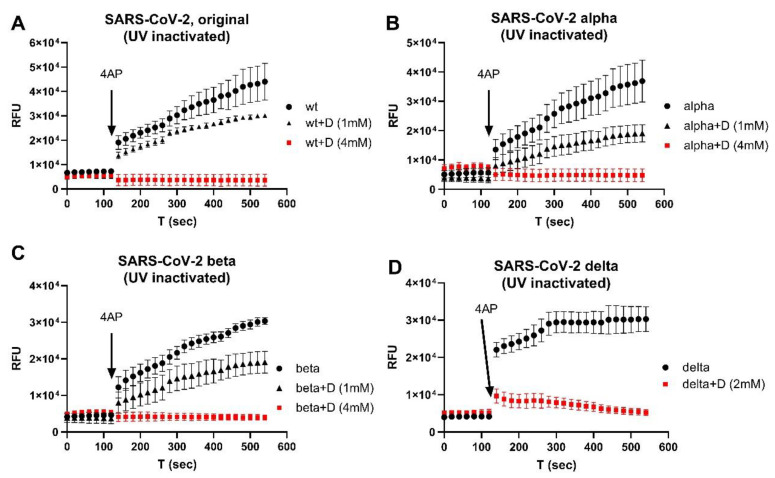
E/Orf3a channel blockers (gliclazide + emodin) on isolated SARS-CoV-2 virus variants: (**A**) original; (**B**) Alpha; (**C**) Beta; (**D**) Delta. All viruses were UV-inactivated. +D: drugs (gliclazide + emodin). The error bars are the average result from 3–6 repeats.

**Figure 3 biomolecules-12-01673-f003:**
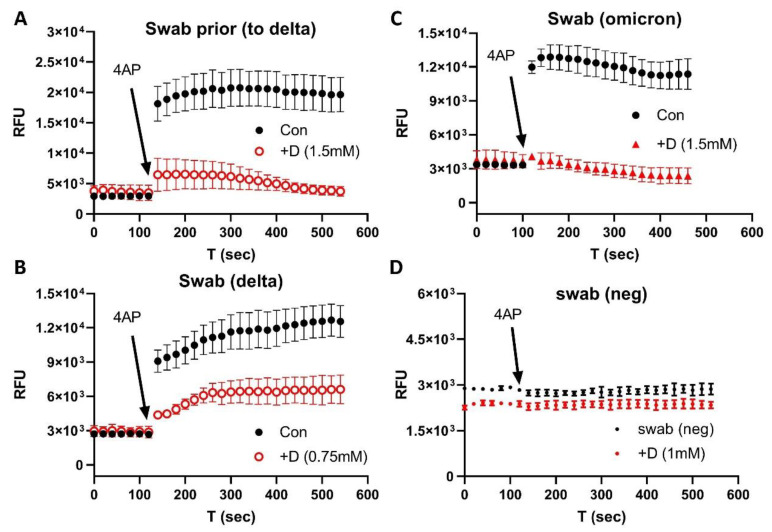
E/Orf3a channel blockers (gliclazide + emodin) on COVID-19 nasal swabs of various variants: (**A**) prior to Delta variant; (**B**) Delta variant; (**C**) Omicron variant; (**D**) negative swab control. Note that the *y*-axis scale is enlarged to show error bars and the drug trace has been moved down to avoid an overlap. +D: gliclazide + emodin. The error bars are the average result from 3–6 repeats.

**Figure 4 biomolecules-12-01673-f004:**
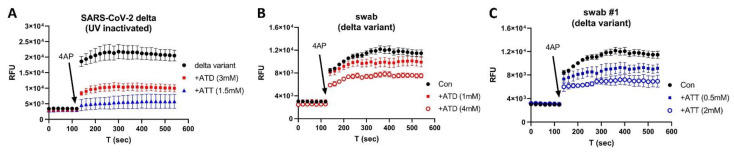
Inhibition of amantadine (ATD) and amitriptyline (ATT) on Delta variant swab: (**A**) isolated Delta virus (UN-inactivated); (**B**) inhibitory effects of ATD on Delta variant swab; (**C**) inhibitory effects of ATT on Delta variant swab.

**Figure 5 biomolecules-12-01673-f005:**
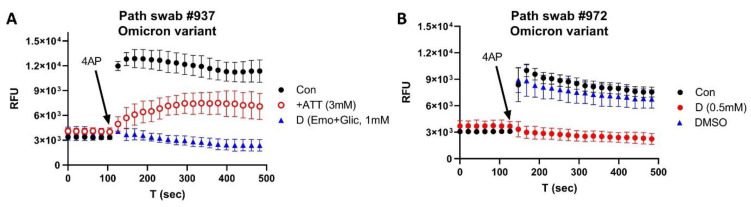
Inhibition of ATT on Omicron variant swab: (**A**) effects of ATT (red) and emodin/gliclazide (blue); (**B**) effects of emodin/gliclazide (red) and DMSO (blue).

**Figure 6 biomolecules-12-01673-f006:**
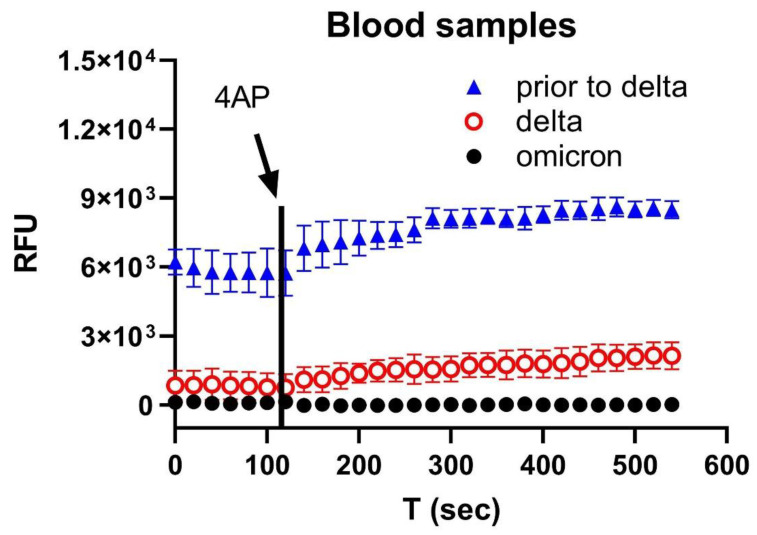
Orf3a/E channel activity of variants in blood. In blood, the SARS-CoV-2 virus channel activity was high in the sample prior to Delta (blue triangle), low during Delta (red circle), and undetectable in Omicron (black circle).

**Table 1 biomolecules-12-01673-t001:** Mutations in the Orf3a and E channel proteins in the variants of concern (VOC).

VOC	Orf3a	Envelope	Channel Activity
Alpha [20]	G254 *	None	(-)
Delta (B.1.617.2) [21]	S26 L	None	(-)
Omicron (BA.1–BA.5) [22,23]	T223 I	T9 I	(-)

(-): not affected because the mutations occur outside the motif that controls the channel activity. E channel activity is controlled by a transmembrane domain between G10 and L37, which forms an ion channel [24,25]. Orf3a K^+^ channel activity is controlled by Y91, Y93, and Y109 [26]. *—mutation.

## Data Availability

The data presented in this study will be openly available after publication at the journal website.

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
