# Peer review of "Inhibition of SARS-CoV-2 Viral Channel Activity Using FDA-Approved Channel Modulators Independent of Variants"

_biomolecules, 2022, doi:10.3390/biom12111673_

Round 1

Reviewer 1 Report

Authors investigated the inhibitory effect of some drugs on several variant of ORF3A/E potassium channel of SARS-CoV-2 virus. Activity of the channels was investigated thanks to a fluorescent Tl+ sensitive probe (Tl+ flow through viroporin potassium channels) as previously described. Experiments were performed on blood and nasal swab. emodin and gliclazide inhibited the channels regardless of the variant in blood sample or nasal swab samples. The drugs amitriptyline and amantadine inhibited also both channels

Main points

In the introduction, authors should explain how the 4-AP activate channels. This is to avoid a confusion as 4-AP is an inhibitor of mammalian potassium channels

Authors wrote: “This technique allows rapid determination of K+ channel activity in a high-throughput manner”.  Unfortunately, authors performed experiment with only two new drugs. The sentence should be modified.

In nasal swab sample, there is virus but also several other type of cells or vesicle of membrane.  It is difficult to understand that the fluorescence is only du to activation of viroporins. Complementary experiences should be performed to confirm that the fluorescence is only due to flow of Ti through the viroporin. Does the fluorescent probe is only in present in the virus and not outside. Is there a direct interaction between the probe and 4-AP?… In the same way, the technic should be better described: quantity of nasal swab sample, estimated concentration of virus, time of incubation of drugs etc..

The drugs concentration used are very high. In previous publication (PMID: 32828269) 50 µM gliclazide have an action on the channels. In the same way amantadine and amitriptyline concentration are high. At this concentration, several side effects could occur as modification of pH and osmolality. This point should be addressed.

Abstract: authors wrote: “they inhibit Orf3a/E channel activity in isolated virus”. Where are experiment on isolated virus?

Minor points

Figure 1 : where are the error bars

Out of curiosity: Gliclazide is an inhibitor of K-ATP. Does K-ATP opener (cromakalim) has the same effect?

Reviewer 2 Report

A fluorescence potassium assay is used to investigate potential blockage of SARS-CoV-2 channel proteins 3a and E protein by known viral channel blockers amantadine and an antidepressant amitriptyline. The screening is validated by analyzing the response of the test by adding known ion channel modulators such as emodin and gliclazide.

It is important to have tests available to screen for novel and repurposing drugs. The results presented seem to be promising.

Some more information about the test assay would be important to evaluate its quality:

It is not clear how the isolated viruses from CDC had been inserted into cells. This would be a prerequisite to undertake the assay. Is this a cell-based assay?

It would be important to show how the IC50 values for amantadine and amitriptyline were obtained.

 When discussing the higher potency of amitriptyline than amantadine (Discussion section), it is stated that it might be due to the effect of the drug on proteins involved in the sodium inward current? Has this affect been blocked in the assay?

In the discussion section 3a and E proteins are described. This could rather move to the introduction in as much they are the main target of the study.

Round 2

Reviewer 1 Report

The authors improved the quality of the manuscript